# Sleep quality and the evolution of the COVID-19 pandemic in five European countries

**Samira Barbara Jabakhanji[1,2], Anthony Lepinteur[3], Giorgia Menta[4] \*, Alan Piper[5], Claus Vögele[3]**

**1** Healthcare Outcomes Research Centre, RCSI University of Medicine and Health Sciences, Dublin, Ireland, **2** Medical Faculty Mannheim, Heidelberg University, Heidelberg, Germany, **3** University of Luxembourg, Esch-sur-Alzette, Luxembourg, **4** Luxembourg Institute of Socio-Economic Research (LISER), Esch-sur-Alzette, Luxembourg, **5** University of Leeds, Leeds, United Kingdom

\* giorgia.menta@liser.lu

## Abstract

The COVID-19 pandemic has led to lifestyle changes across Europe with a likely impact on sleep quality. This investigation considers sleep quality in relation to the evolution of the COVID-19 pandemic in five European countries. Using panel regressions and keeping policy responses to COVID-19 constant, we show that an increase in the four-week average daily COVID-19 deaths/100,000 inhabitants (our proxy for the evolution of the pandemic) significantly reduced sleep quality in France, Germany, Italy, Spain, and Sweden between April 2020 and June 2021. Our results are robust to a battery of sensitivity tests and are larger for women, parents and young adults. Additionally, we show that about half of the reduction in sleep quality caused by the evolution of the pandemic can be attributed to changes in lifestyles, worsened mental health and negative attitudes toward COVID-19 and its management (lower degree of confidence in government, greater fear of being infected). In contrast, changes in one's own infection-status from the SARS-CoV-2 virus or sleep duration are not significant mediators of the relationship between COVID-19-related deaths and sleep quality.

## 1. Introduction

The COVID-19 pandemic has changed many aspects of life worldwide, some of which are associated with individual sleep quality and our overall wellbeing. The pandemic has, for example, made labour markets more insecure [1–3] and lifestyles more sedentary [4–7]. Furthermore, the growing body of literature on COVID-19 has shown that the pandemic is also responsible for a rise in mental-health issues [8–10] and loneliness [11–13], among other things. We here contribute to this literature by assessing how the evolution of the pandemic affected sleep quality in five European countries, net of the effects of national pandemic policies.

Sleep affects a wide range of outcomes, including COVID-19-related outcomes. Previous studies show that poor sleep can have detrimental consequences on subjective well-being [14], labour market outcomes [15, 16], decision-making [17], cognitive and motor performance

**Data Availability Statement:** The COME-HERE data contain sensitive individual information and cannot be made publicly available (Ethic Research Panel of the University of Luxembourg, decision number ERP 20-026 C/A COME-HERE). Data

access to the University of Luxembourg server can be obtained on request for noncommercial research purposes (contact persons: Conchita D'Ambrosio, conchita.dambrosio@uni.lu; Claus Vögele, claus.voegele@uni.lu).

**Funding:** Data collection was possible thanks to funding from the André Losch Fondation (https://www.loschfondation.lu), Art2Cure (https://www.bil.com/galerie-lindependance/art2cure/index.html), Cargolux (https://www.cargolux.com/), CINVEN Foundation (https://www.cinven.com/) and COVID-19 Foundation (https://www.fdlux.lu/en/node/1643), under the aegis of the Fondation de Luxembourg, Fonds National de la Recherche Luxembourg (14840950 –COME-HERE, https://www.fnr.lu/) and the French Agence Nationale de la Recherche (ANR-21-CO16-0002, https://anr.fr/projet-ANR-21-CO16-0002). The funders had no role in study design, data collection and analysis, decision to publish, or preparation of the manuscript.

**Competing interests:** The authors have declared that no competing interests exist.

[18–20], metabolism [21] and cardiovascular health [22]. [23] also find that poor sleep in the weeks before exposure to the common cold (a rhinovirus) was associated with a greater probability to develop the illness, implying a protective role of sleep on immunity. Similarly, [24] suggest that a lack of sleep may have contributed to the spread of COVID-19 by compromising the immune system. Additionally, they find that a lack of sleep can act as a barrier to psychological functioning and decision-making which may, in turn, affect compliance with recommendations set out to combat COVID-19 transmission. In a similar vein, [25] have shown that sleep quality plays a role in COVID-19 vaccine efficacy. Good sleep therefore appears crucial to maintain good health and wellbeing, particularly during health-related crises such as the COVID-19 pandemic; an observation that will also hold for future health crises and pandemics.

There is reason to believe that the pandemic itself affected our sleep quality, as a number of studies show a high prevalence of poor sleep quality during the pandemic. Using a survey collected via Facebook in Jordan in March 2020, [26] find that over half of the 1260 participants had recently had sleep problems. [27, 28] find similar results in Italy and China, respectively. Other studies reported poorer sleep quality during the pandemic among students [29, 30], late adolescents and young adults [31], health professionals [32], COVID-19 patients [33], and men [34].

Multiple potential pathways might explain how the pandemic affected sleep quality. First, catching the COVID-19 virus can affect sleep quality directly through physiological mechanisms. Second, irrespective of being personally affected by the virus, the pandemic might have affected sleep quality indirectly, by inducing changes in sleep-related behaviour. Several authors document a higher adoption of sedentary lifestyles during the pandemic [4–7], as well as overall worse mental health [35, 36], both factors known to be important determinants of sleep quality long before the outbreak of COVID-19 [37–39]. Sleep duration might also have changed and affected sleep quality. Concerns and negative attitudes about the COVID-19 crisis and its management may also be potential indirect pathways: the sleep quality of individuals suffering from the so-called "coronaphobia" [40], the fear of the coronavirus, or reporting lower level of trust in government responses [41] might have deteriorated. It is unclear to which extent these pathways affected pandemic sleep quality across Europe, particularly during peak infection times, and whether some population groups were more affected than others.

Prior to the pandemic, sleep quality has been found to depend on individual characteristics. In general, studies find more problems with sleep for females than males [42, 43], although there is some evidence that the patterns by gender may be more nuanced [44]. Parents, and particularly first-time parents, report reduced sleep quality, a result more pronounced for women, whose sleep quality already suffers during pregnancy [45, 46], as compared to men [47, 48]. The pre-pandemic literature also indicates that sleep patterns can differ based on age, education and income. Sleep quality has been found to have an association with age, with one study from Germany finding a decline from age 18 until age 60 [49]. These results are underpinned by the meta-analysis by [50], who document a similar decline in sleep quality from childhood until old age. Other research found that sleep quality tends to increase with both higher income status and education [51]. As shown by [52], these relationships appear to hold in middle-income countries as well as in the more researched high-income countries. Overall, as various population groups appeared more prone to poor sleep quality than others before the pandemic, we expect the effect of the pandemic on sleep quality to be heterogeneous.

Adding to the prior research, we present new evidence on sleep quality during the pandemic in five European countries. Using the longitudinal COME-HERE survey (COVID-19, MEntal HEalth, REsilience and self-regulation), we assess the influence of the evolution of the

COVID-19 pandemic on respondents' sleep quality using linear panel regressions. Following [53], we take the four-week average number of daily deaths per 100,000 inhabitants as a proxy of the evolution of the pandemic in each country. The results suggest a negative impact of the pandemic on sleep quality: a one standard-deviation increase in the daily death rate was associated with a significant reduction of 3 percent of standard deviation in sleep quality, net of the effect of pandemic policies. This is comparable in absolute terms to about 40% of the sleep-quality premium of doubling the pre-pandemic household income. Consistent with the literature on sleep-quality heterogeneity described above, we find the association between sleep quality and the evolution of the pandemic to be larger for certain sub-groups of the population (women, the younger, and parents). We additionally investigate some of the potential mechanisms underlying our main results. Using a decomposition approach [54], we show that the reduction in sleep quality caused by the evolution of the pandemic is mostly explained by changes in lifestyles, worsened mental health and negative attitudes toward COVID-19. We do not find any evidence that the effect of the evolution of the pandemic is mediated by having been infected by the SARS-CoV-2 virus. We also show that sleep duration (as proxied by the time spent in bed) increases with the intensity of the pandemic. Nevertheless, the rise in sleep duration is small: a one standard-deviation increase in the four-week average number of daily deaths is associated with 3 additional minutes in bed per night. Our mediation analysis shows that this small rise in sleep duration has no effect on sleep quality.

We contribute to the literature assessing the effect of the COVID-19 pandemic in several aspects. First, the extant studies typically involve relatively small and non-representative samples in cross-sectional surveys. The database we use here, namely the COME-HERE survey, overcomes these limitations, providing nationally-representative data for France, Germany, Italy, Spain and Sweden at six time points between April 2020 and June 2021. Second, the longitudinal design of the survey offers several advantages. We can follow the same individuals through different moments of the pandemic, starting from as early as the first European lockdowns. We can also implement panel regression analyses and thus keep constant the influence of unobserved time-invariant heterogeneity. (This is particularly important in a sleep context where individuals can have quite heterogenous sleep habits. Our longitudinal approach thus constitutes an advantage as compared to cross-sectional studies that cannot rule out capturing someone's 'natural' sleep behaviour or sleep needs in their analyses). In a context where finding purely exogenous variations in the intensity of the pandemic is almost impossible, being able to condition on both individual unobserved heterogeneity and the daily changes in national governments' policy responses produces estimates that can arguably be read as causal.

The remainder of the paper is organised as follows. Section 2 describes the data and the empirical strategy. The main results, the robustness checks and the heterogeneity analysis are reported in Section 3. Section 4 presents the mechanisms and Section 5 concludes.

## 2. Data and empirical strategy

### 2.1. Data

The data we use here come from the ongoing COME-HERE survey collected by the University of Luxembourg. The survey was conducted online through the Qualtrics platform to produce nationally representative samples of adults (aged 18 years and over) in France, Germany, Italy, Spain and Sweden. Sample stratification ensured that the data is representative in terms of gender, region, and age. Ethics approval was granted by the Ethics Review Panel of the University of Luxembourg (approval number: ERP 20–026 C/A COME-HERE). Respondents were asked to complete an online questionnaire that takes approximately 20 minutes, collecting information both at the individual and household level. Informed written consent was obtained from

all participants for the collection and use of their responses for research purposes. The nature of the survey is longitudinal: respondents were first interviewed around the end of April 2020, and then re-contacted for additional survey waves in early June 2020, early August 2020, late November 2020, March 2021, and June 2021. Additional waves that took place between the end of 2021 and early 2022 were not yet available at the time of analysis.

More than 8,000 individuals responded to the first survey wave, and were then invited to take part in the subsequent waves. Over 75% of wave-one respondents participated in at least one other survey wave, with 34% participating in all six. See S1 Fig for further details on the survey timeline and the number of respondents per wave. The survey contains cross-sectional weights guaranteeing the national representativeness of the samples over time, as well as longitudinal Inverse-Probability Weights addressing the issue of non-random attrition. The survey collects detailed information on individuals' living conditions, lifestyles and physical and mental health during the pandemic; it also identifies recent changes and events in their lives. Standard sociodemographic characteristics presented in the literature review above, such as age, gender, education, labour-force status, and country and region of residence, are also included in the survey.

In each survey round, respondents replied to the following question to assess sleep quality, "*How would you rate your sleep quality during the last week on average*?", using a standard 7-point Likert scale ranging from "very poor" to "excellent". This single-item measure is similar to the most direct question about sleep quality asked as part of the validated and extensively used multi-item Pittsburgh Sleep Quality Index (PSQI) [55, 56].

## 2.2. Empirical strategy

Using the data described above, we here look at how the evolution of the COVID-19 pandemic, net of national pandemic policy measures, affected respondents' sleep quality. In order to do so, we estimate the following equation via OLS with individual fixed-effects:

$$SQ_{ijt} = \alpha COVID_{jt} + \beta X_{ijt} + \delta Policy_{jt} + \mu_i + \lambda_t + \varepsilon_{ijt}. \tag{1}$$

Here $SQ_{ijt}$ is the sleep quality of respondent $i$ living in country $j$ at time $t$. $COVID_{jt}$ is the four-week average daily deaths per 100,000 inhabitants prior to the interview date of the COME-HERE respondents. As shown in [53], this measure is arguably the most accurate proxy of the evolution of the pandemic as it is the best predictor of governments' policy reactions (such as lockdowns), performing better than alternative metrics based on number of infections.

The vector $X_{it}$ includes individual characteristics that are traditionally used in sleep-quality regressions, namely age categories, the log of the monthly disposable household income, (Monthly disposable household income was equivalised using a square-root equivalence scale in order to account for within-household economies of scales. Additionally, income is adjusted for a purchasing power parity (PPP) exchange rate.) and dummies for gender, partnership status, parenthood, education (lower-, upper-, and post-secondary), employment status, population density of the place of residence, and country of residence [14, 57]. We control for macro-trends and individual time-invariant heterogeneity by introducing, respectively, wave fixed-effects $\lambda_t$ and individual fixed-effects. In our panel estimations, all of the $X_{it}$ variables other than income and employment status will be dropped due to their time-invariant nature in the survey. Standard errors are clustered at the individual level and we do not weight our observations in the main specification. We present a number of robustness checks in section 3.2 to show that our conclusions hold with different alterations of our main specification.

Since the four-week average daily deaths per 100,000 inhabitants predicts policy responses to the pandemic, $\alpha$ may confound the effect of the pandemic itself with that of the pandemic

policies. This is why we control for the vector $Policy_{jt}$. It contains the two-week average Stringency Index and the two-week average Economic Support Index produced by the Blavatnik School of Government at the University of Oxford, as part of the Oxford COVID-19 Government Response Tracker [58]. Over one hundred international students and staff members at the University of Oxford collect data from public sources to produce indices measuring policy responses to COVID-19 at the national level that are updated on a daily basis (The Oxford COVID-19 Government Response Tracker does not currently have international data on the level of regional policies). The Stringency Index is composed of the nine following sub-indices, measuring various aspects of containment policies: "school closing", "workplace closing", "cancellation of public events", "restriction on gathering", "public transport closing", "stay-at-home requirements", "restriction on internal movement", "restriction on international travel" and "public information campaign". Additionally, the Economic Support Index has two components: "income support" and "debt relief"."Income support" measures the extent to which governments provide their citizens with direct cash payments, universal basic income, or income support for those who lost their job or cannot work;"debt relief" pertains to governmental decisions to freeze the financial obligations of households (such as loan repayments). (For more details, see www.bsg.ox.ac.uk/research/research-projects/coronavirus-government-response-tracker#data).

We expect $\alpha$ to be negative: as the pandemic becomes more lethal, sleep quality should worsen, due to the direct and/or indirect mechanisms described in the literature above. We also believe that our empirical model produces a coefficient $\alpha$ that is arguably causal for several reasons. First, our model does not suffer from reverse causation: there are no reasons to believe that the sleep quality of an individual at interview date $t$ will influence the daily COVID-19 deaths. This is even less plausible provided that we use the four-week average daily deaths prior the interview date $t$. Second, it could be argued that $\alpha$ may capture the influence of confounders. We attenuate this threat as much as possible by controlling for a rich set of individual characteristics, individual fixed-effects and a vector of variables aiming at keeping the influence of pandemic policies constant.

To estimate our model, we consider the sample of COME-HERE respondents who were present in at least two out of the first six survey waves, and who provided valid information on sleep quality and the socio-demographic variables used as controls. This sample consists of 27,728 observations (for 6,190 individuals) observed between April 2020 and June 2021; the associated descriptive statistics appear in Table 1. French, German, Italian and Spanish respondents make up for little over 20% of the sample each, while 12% of the observations come from Swedish respondents. In terms of the wave structure, 20% of the observations are from wave one, and the remainder are fairly equally distributed across the five remaining waves. Just under half of the sample observations come from women and the highly educated (*i.e.* those with at least a diploma from post-secondary education). As with all panel surveys, there is attrition. While we do not use weights in our main specification, in Section 3.2 we show that our results are robust to the use of longitudinal weights accounting for non-random attrition.

Fig 1 shows the distribution of sleep quality in the estimation sample. About 2.5% of respondents reported having "very poor" sleep in the week before the interview, while 9% reported their sleep being "excellent". This left-skewed distribution is common to other sleep quality studies (for example Talamini et al. 2013) and equivalent to the right-skewed distribution of studies using the PSQI [59, 60] (where higher numbers are associated with poor quality sleep) indicating that, in general, most people enjoy at least a reasonable level of sleep quality. By pooling individual observed between April 2020 and June 2021, one may argue that Fig 1 potentially hide shifts in the distribution of sleep quality over time. This is not confirmed by

**Table 1. Descriptive statistics–estimation sample.**

|  | Mean | SD | Min | Max |
|---|---|---|---|---|
| Time in Bed | 483.67 | 81.76 | 180 | 840 |
| Sleep Quality | 4.70 | 1.44 | 1 | 7 |
| Average Daily Deaths/100,000 inhabitants (4-week average) | 0.39 | 0.34 | 0 | 1.26 |
| Stringency Index (2-week average) | 69.17 | 11.84 | 46.30 | 93.52 |
| Economic Support Index (2-week average) | 65.48 | 19.65 | 29.69 | 100 |
| France | 0.22 |  | 0 | 1 |
| Germany | 0.21 |  | 0 | 1 |
| Italy | 0.22 |  | 0 | 1 |
| Spain | 0.23 |  | 0 | 1 |
| Sweden | 0.12 |  | 0 | 1 |
| Wave One | 0.20 |  | 0 | 1 |
| Wave Two | 0.15 |  | 0 | 1 |
| Wave Three | 0.18 |  | 0 | 1 |
| Wave Four | 0.18 |  | 0 | 1 |
| Wave Five | 0.16 |  | 0 | 1 |
| Wave Six | 0.14 |  | 0 | 1 |
| Age: 18 to 29 years | 0.13 |  | 0 | 1 |
| Age: 30 to 39 years | 0.17 |  | 0 | 1 |
| Age: 40 to 49 years | 0.19 |  | 0 | 1 |
| Age: 50 to 59 years | 0.18 |  | 0 | 1 |
| Age: 60 to 69 years | 0.23 |  | 0 | 1 |
| Age: 70+ years | 0.09 |  | 0 | 1 |
| Female | 0.49 |  | 0 | 1 |
| Lower-Secondary Education | 0.19 |  | 0 | 1 |
| Upper-Secondary Education | 0.38 |  | 0 | 1 |
| Post-Secondary Education | 0.43 |  | 0 | 1 |
| Living with a Partner | 0.62 |  | 0 | 1 |
| Children at Home | 0.33 |  | 0 | 1 |
| Log of Household Equivalised Income (in PPP) | 7.26 | 0.67 | 4.94 | 9.42 |
| Employed | 0.58 |  | 0 | 1 |
| Pop. Density: Isolated Dwelling | 0.02 |  | 0 | 1 |
| Pop. Density: Less than 2,000 | 0.09 |  | 0 | 1 |
| Pop. Density: Between 2,000 and 10,000 | 0.18 |  | 0 | 1 |
| Pop. Density: Between 10,000 and 50,000 | 0.21 |  | 0 | 1 |
| Pop. Density: Between 50,000 and 100,000 | 0.14 |  | 0 | 1 |
| Pop. Density: More than 100,000 | 0.37 |  | 0 | 1 |
| *Mediators* |  |  |  |  |
| Infected by SARS-CoV-2 | 0.02 |  | 0 | 1 |
| Days outside per week | 4.63 | 2.31 | 0 | 7 |
| Days with phyiscal activity per week | 2.55 | 2.25 | 0 | 7 |
| Daily working time (in hours) | 3.82 | 4.30 | 0 | 23.5 |
| Anxiety (GAD-7 scale) | 5.34 | 5.23 | 0 | 21 |
| Depression (PHQ-9 scale) | 6.06 | 6.13 | 0 | 27 |
| Confidence in the government | 4.34 | 1.83 | 1 | 7 |
| Worry about getting seriously ill from SARS-CoV-2 | 2.88 | 1.54 | 1 | 6 |

*Note*: These numbers refer to respondents from our estimation sample (27,728 observations) coming from the four 2020 waves and the first two 2021 waves of the COME-HERE survey.

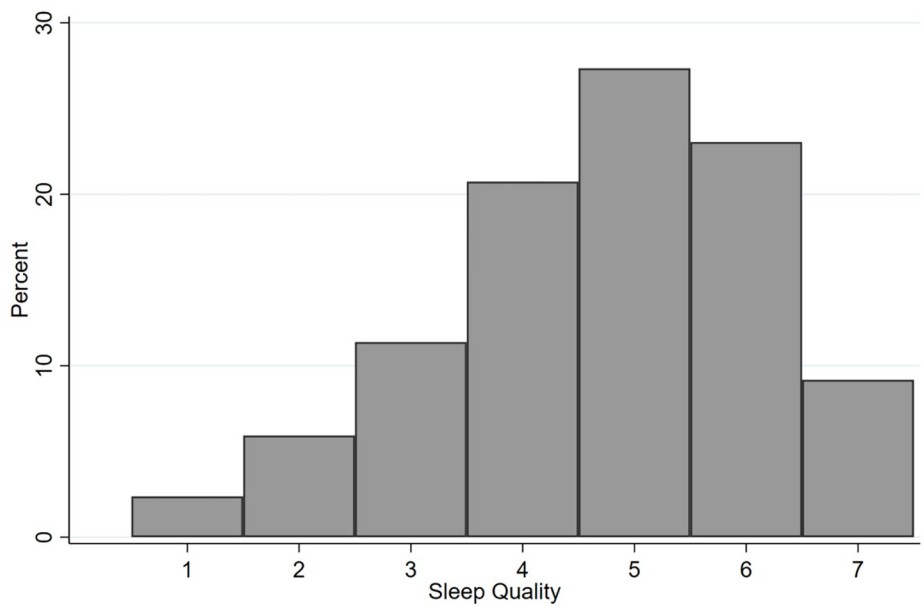

**Fig 1. Distribution of sleep quality–estimation sample.** *Note*: These numbers refer to respondents from our estimation sample (28,572 observations) coming from the four 2020 waves and the first two 2021 waves of the COME-HERE survey.

our data: the distribution of sleep quality displays the same features in all periods covered by our dataset (figures available upon request).

## 3. Sleep quality and the pandemic

### 3.1. Main results

Table 2 lists the regression results from the estimation of Eq (1). Column (1) introduces the four-week average daily death in our sleep-quality regression with country and wave fixed-effects for only controls. As we expected, the estimate attracted is negative and highly

**Table 2. Sleep quality and COVID-19 deaths–pooled and panel results.**

|  | Sleep Quality (1–7 –standardised) | | | |
|  | **(1)** | **(2)** | **(3)** | **(4)** |
| --- | --- | --- | --- | --- |
| Average Daily Deaths/100,000 inhabitants | -0.026*** | -0.036*** | -0.035*** | -0.033*** |
| (4-week average) | (0.010) | (0.010) | (0.008) | (0.010) |
| Observations | 27728 | 27728 | 27728 | 27728 |
| Individual Controls | No | Yes | Yes | Yes |
| Individual FE | No | No | Yes | Yes |
| Pandemic Policies | No | No | No | Yes |

*Notes*: These are linear regressions. The sample here is respondents coming from the four 2020 waves and the first two 2021 waves of the COME-HERE survey. Sleep quality and the average daily deaths variable are standardised over the estimation sample. Standard errors in parentheses are clustered at the individual level. The individual controls are age categories, gender, education, parenthood, relationship status, and population density (all measured at Wave 1), the log of equivalent household disposable income in PPP, and a dummy for the employment status. The pandemic policies are the two-week averages of the Stringency Index and Economic Support Index. All regressions include wave and country fixed-effects

*, **, and *** respectively indicate significance levels of 10%, 5% and 1%.

significant: the more lethal the pandemic was between April 2020 and June 2021, the lower the sleep quality of the respondents of our estimation sample.

We then introduce a vector of individual controls in column (2) and the individual fixed-effects in column (3). We find estimates that are still negative, significantly different from zero at conventional levels but not different from that of the first column, suggesting that the relationship between the evolution of the pandemic and sleep quality is orthogonal to other individual characteristics. Last, we control for the influence of pandemic policies in the last column of Table 2 by controlling for the average values of the Stringency Index and the Economic Support Index in the two weeks prior to respondents' interview dates. As a result, a one standard-deviation increase in the four-week average daily deaths per 100,000 inhabitants is associated with a statistically significant reduction of 3.2% of a standard deviation in sleep quality. This is a sizeable effect: according to S1 Table, where we report the point estimates and standard errors for all the control variables, it is similar to the effect of doubling income in absolute terms.

## 3.2. Robustness checks

We report results from a battery of robustness checks in S2 Table. Column (1) reproduces the benchmark estimates already reported in the last column of Table 2, in order to ease the comparisons across specifications.

We first ask whether our conclusions are affected by the way we treat our dependent variables. Our baseline specification treats sleep quality as a cardinal variable. However, as sleep quality is recorded in the survey as an ordinal variable, non-linear models may be a more appropriate tool to frame our research question. This is why we report the coefficient for the average daily COVID-19 deaths per 100,000 inhabitants coming from a fixed-effects ordered logit, using the 'Blow-Up and Cluster' (BUC) estimator of [61] in column (2). Although this coefficient is not a marginal effect, its sign and significance can be interpreted: using a non-linear model produces conclusions that are qualitatively similar to that of main regressions.

We then transform our dependent variable into a dummy indicating a high sleep quality (above the median) and report the results in column (3) of S2 Table. Consistent with the main specification, results here indicate that a one standard-deviation increase in the four-week average number of daily deaths reduces by 2 percentage points the probability of reporting higher sleep quality than the median person in the sample. (Similar results hold when using a conditional logit model instead of a linear probability model (results not shown).

We then turn to the right-hand side of the specification and show that our results do not change when measuring the pandemic's evolution using the two-week average number of daily deaths per 100,000 inhabitants, instead of the four-weeks average used at baseline (see column 4).

Last, we test whether our results are sensitive to the clustering level of standard errors and to the introduction of cross-sectional or longitudinal weights. In column (5) of S2 Table, we find almost identical standard errors when clustering at the $COVID_{jt}$ level (that is, the level of variation of the independent variable, see [62]). We then apply longitudinal and cross-sectional weights, respectively, in columns (6) and (7) of the same Table to address concerns about representativeness and attrition in COME-HERE. Results are once again qualitatively the same across specifications: a greater COVID-19 mortality is always associated with significant reductions in sleep quality.

## 3.3. Heterogeneity

COVID-19 mortality might affect the sleep quality of some respondents more than others, based on characteristics that might make them more or less vulnerable to the adverse effects of

the pandemic on their sleep patterns. We do not know *a priori* whether the pandemic could have affected disproportionately more those who already reported poorer sleep quality (e.g., women, parents) or, on the contrary, those with a better average sleep quality to begin with. To address this question, we interact the four-week average daily deaths with a set of relevant socio-economic characteristics (gender, age, education, parenthood, income) and report the results in S3 Table. Column (1) shows that the reduction in sleep quality is twice as large for women than for men, and that the difference between gender is significant at the 1% level. We also observe a significant difference between respondents above and below the median age of 51 years in column (2): a one-standard deviation increase in the average daily COVID-19 deaths reduces older respondents' sleep quality by 2% of a standard deviation, while it reduces that of younger respondents by 4% of a standard deviation. We find no differences based on the level of education in column (3), but the next column reveals that the effect of the mortality of the pandemic is significantly larger for parents than respondents without children. (Respondents were asked "Are there children living with you in your household?" which we use as a proxy variable indicating parenthood. We acknowledge that a minority of respondents who answered "yes" may not be the parent of those children living in the same household, and that some parents may be living apart from their children thus replying "no"). Similar to results on education, we find no protective effect of household income on sleep quality in the last column of S3 Table.

Results suggest that the pandemic had an inequality-enhancing effect on sleep quality for women and parents, both groups that other studies have shown to more frequently experience poor sleep quality [42, 43, 47, 48]. Conversely, the pandemic reduced age-based inequalities in sleep quality–the disproportionately higher fall in young people's sleep quality narrowing the gap between the young and the old.

## 4. Mechanisms

Why did sleep quality decrease with the increased severity of the mortality rate of the COVID-19 pandemic? A major determinant of sleep quality is sleep duration [63]. While sleep duration is not directly reported in COME-HERE, we use the time spent in bed as a plausible proxy. Specifically, respondents were asked to report the average time at which they went to bed and at which they woke up during the week before the interview, the difference of which we take as time spent in bed.

We replicate our main analysis in Table 3, using the time in bed as a dependent variable. Our estimates do not suggest that the loss in sleep quality we observed earlier is caused by changes in the time spent in bed. As the pandemic worsens (as measured by our daily deaths proxy), time in bed increases. This could be explained by the reduced availability of alternative activities (such as work/commuting or outdoor leisure) during the hardest moments of the pandemic, potentially leading to longer time spent in bed due to boredom or reduced opportunity cost of time now spent sleeping. However, in contrast to our sleep quality results, the statistical significance and the precision of estimates do not imply economic meaningfulness as effect sizes are extremely low: a one standard-deviation increase in the four-week average daily COVID-19 deaths increases the time spent in bed by 3 minutes on average. The average time spent in bed in the estimation sample is about 484 minutes (8.07 hours) per day, and 3 minutes equate to a 0.6% increase in the daily average (Table 1). Overall, our results do not indicate that the degradation in sleep quality caused by the evolution of the pandemic can be explained by a reduction in sleep duration; thus, we explore other factors as follows.

Sleep quality depends on factors that were themselves likely to be affected by the evolution of the pandemic. First, the adoption of more sedentary lifestyles that accompanied the

**Table 3. Sleep duration and COVID-19 deaths–pooled and panel results.**

|  | Time in Bed (in minutes) | | | |
|---|---|---|---|---|
|  | (1) | (2) | (3) | (4) |
| **Average Daily Deaths/100,000 inhabitants** | 3.298*** | 2.914*** | 2.998*** | 3.306*** |
| **(4-week average)** | **(0.818)** | **(0.810)** | **(0.743)** | **(0.859)** |
| Observations | 27728 | 27728 | 27728 | 27728 |
| Controls | No | Yes | Yes | Yes |
| Individual FE | No | No | Yes | Yes |
| Pandemic Policies | No | No | No | Yes |

*Notes*: These are linear regressions. The sample here is respondents coming from the four 2020 waves and the first two 2021 waves of the COME-HERE survey. The average daily deaths variable are standardised over the estimation sample. Standard errors in parentheses are clustered at the individual level. The individual controls are age categories, gender, education, parenthood, relationship status, and population density (all measured at Wave 1), the log of equivalent household disposable income in PPP, and a dummy for the employment status. The pandemic policies are the two-week averages of the Stringency Index and Economic Support Index. All regressions include wave and country fixed-effects

*, **, and *** respectively indicate significance levels of 10%, 5% and 1%.

COVID-19 outbreak influenced the time spent performing daily activities (less time outside, fewer opportunities for leisure, reduced working hours, etc.). These changes may in turn affect sleep quality [39, 64]. Second, the rise of mental health issues, such as depression, during the pandemic [8] may contribute to explaining our main results. We also postulate that changes in attitudes specifically related to COVID-19 (such as the confidence in the government to handle the crisis, or the fear of infection) may contribute to a reduction in sleep quality. As the pandemic progressed, differences in public trust towards various government responses were observed between countries and over time [41]. Last, as the pandemic progressed, the individual probability of having become infected with the SARS-CoV-2 virus increased, with potential long-term consequences on respondents' sleep quality.

To assess whether these potential channels mediate the baseline results reported in Table 2, we follow the decomposition approach developed by [54]. The decomposition relies on the omitted-variables bias formula and can be used to attribute a portion of an estimate to potential groups of mediators. COME-HERE respondents were asked to report the average number of hours they spent working on an average working day of the week before the interview, as well as the numbers of days in which they went outside and during which they performed moderate or vigorous physical activity for at least 15 minutes in the week preceding the interview. We use these variables to capture the effect of changes in lifestyles. Additionally, we measure the influence of changes in mental health using the validated psychometric Generalised Anxiety Disorder assessment (GAD-7) and the Patient Health Questionnaire (PHQ-9), scales that measure anxiety and depression symptoms respectively. Response categories for both the GAD-7 and PHQ-9 items were 'Not at all', 'Several days', 'More than half the days' and 'Nearly every day'. Individual scores were converted and summed to a continuous composite score for GAD-7 and PHQ-9, each, following standard procedures. Attitudes towards COVID-19 and national governments' responses to the crisis are also collected in the survey. Here we rely on the following two questions: the first asks about the degree of confidence in the government's ability to handle the COVID-19 crisis well, and is recorded on a seven-point Likert scale, with values ranging from 1 'None at all' to 7 'Absolutely'; the second measures how worried respondents are about the possibility of becoming seriously ill with COVID-19, and is recorded on a six-point Likert scale, with values ranging from 1 'Never' to 6 'All the time'. Additionally, COME-HERE respondents reported in each wave whether they had ever been infected by the

**Table 4. Sleep quality and COVID-19 deaths–gelbach decomposition.**

| | Sleep Quality (1–7 –standardised) | | |
|---|---|---|---|
| | **Base (1)** | **Full (2)** | **Explained (3)** |
| Average Daily Deaths/100,000 inhabitants | -0.033*** | -0.016* | -0.016*** |
| (4-week average) | (0.010) | (0.009) | (0.003) |
| *Contributions*: | | | |
| Infected by SARS-CoV-2 | | | 0.000 |
| | | | (0.000) |
| Changes in lifestyle | | | -0.006*** |
| | | | (0.001) |
| Changes in GAD-7 and PHQ-9 | | | -0.006*** |
| | | | (0.001) |
| Changes in attitudes and worry towards COVID-19 | | | -0.005** |
| | | | (0.002) |
| Observations | 27728 | 27728 | |
| Controls | Yes | Yes | |
| Individual FE | Yes | Yes | |
| Pandemic Policies | Yes | Yes | |

*Notes*: These are linear regressions based on the decomposition approach of Gelbach (2016). The sample here is respondents coming from the four 2020 waves and the first two 2021 waves of the COME-HERE survey. The average daily deaths variable are standardised over the estimation sample. Standard errors in parentheses are clustered at the individual level. The individual controls are age categories, gender, education, parenthood, relationship status, and population density (all measured at Wave 1), the log of equivalent household disposable income in PPP, and a dummy for the employment status. The pandemic policies are the two-week averages of the Stringency Index and Economic Support Index. All regressions include wave and country fixed-effects

*, **, and *** respectively indicate significance levels of 10%, 5% and 1%.

SARS-CoV-2 virus, which allows us to consider respondents with potential physiological long-term consequences after SARS-CoV-2 infection ('long Covid').

We report the results of the mediation decomposition in Table 4. For ease of comparison, column (1) replicates our baseline estimates. In column (2), we control for all the possible mediators at once. When we do, the estimates attracted by the four-week average daily deaths is divided by two and only significant at the 10% level. It already suggests that changes in lifestyles, mental health and attitudes towards COVID-19 are likely to mutually explain most of the main effect we identified in Table 2. Column (3) disentangles the contribution of each set of mediators, revealing that lifestyles, mental health and attitudes towards COVID-19 equally contribute to the share of the main effect that is explained by the model in column (2). (Mental health here includes both the GAD-7 anxiety scale and the PHQ-9 depression scale. The two are predictably highly correlated (raw correlation of 0.83). However, the mediation analysis does not appear to be affected by the potential multi-collinearity between the two: results still hold with the same magnitude and at conventional significance levels when only including either one of the two measures of mental health in the analysis). Prior COVID-19 infection in turn was not associated with sleep quality, likely because of the low incidence of infections in our sample (less than 1.6%).

We additionally estimate the effect of the four-week average daily COVID-19-deaths on the potential eight mediators in separate regressions, mirroring our baseline specification; results are reported in S4 Table. The estimates are consistent with the conclusions of the decomposition approach. First, the probability to be infected with SARS-CoV-2 is orthogonal to our independent variable and, as such, has no significant influence in the decomposition. The contribution of the change in lifestyles appears to be entirely driven by the reduction in time

spent outside during the pandemic. Both the depression and anxiety scale seem to equally contribute to the significance of the mental health channel. The same applies to the confidence in the government and the degree of worries to become ill with COVID-19 for the attitudes towards COVID-19. Note that the COME-HERE survey includes a variety of questions about respondents' worries (e.g., about family, friends, finances). We do not control simultaneously for all these sources of worry because they are highly correlated. However, we performed a factor analysis to produce a single worry score. When substituting the worry to become ill with COVID-19 with this single score in our Gelbach decomposition, the latter attracts a much smaller coefficient. This means that our main effect is not explained by a general increase in worry but by the increase in the worry about catching COVID-19 only.

## 5. Conclusions

Using longitudinal data from five European countries, we assess the influence of the evolution of the COVID-19 pandemic on sleep quality between April 2020 and June 2021. We show that a one standard-deviation increase in the four-week average number of daily deaths per 100,000 inhabitants (our proxy for the evolution of the pandemic) is associated with a reduction of 3% of a standard deviation in sleep quality, net of the effect of pandemic policies. This result survives a battery of robustness checks. The association between sleep quality and the evolution of the pandemic is larger for some sub-groups of the population (women, parents, and the young), suggesting that pre-existing inequalities in sleep quality were exacerbated for women and parents as a result of the evolution of the pandemic while age-related inequalities shrank. We then provide evidence concerning the mechanisms lying behind this association. Using a descriptive decomposition of the omitted variable bias [54], we show that about half of the reduction in sleep quality caused by the evolution of the pandemic can be attributed to changes in lifestyles, worse mental health and negative attitudes toward COVID-19, but not to changes in one's own infection-status from the SARS-CoV-2 virus. We also show that sleep duration (proxied by the time spent in bed) increases with the intensity of the pandemic. However, the rise in sleep duration is small: a one standard-deviation increase in the four-week average number of daily deaths is associated with a 3-minute increase of time spent in bed per night. This is not enough to counterbalance the negative consequences of changes in lifestyle, mental health and attitudes towards COVID-19 on sleep quality.

The interpretation of our results should be informed by some limitations. While we argue that our identification strategy is sufficient to infer a causal effect of the evolution of the pandemic on sleep quality, we cannot compare our findings to a counterfactual scenario in the absence of COVID-19. Our results may thus be partly confounded by unobservable time-varying characteristics that are both correlated with individual sleep quality and the daily COVID-19 death rate. However, we believe these potential threats to be of second order and thus unlikely to severely affect our results. Lastly, we acknowledge the relatively high attrition rate which is not surprising given the many disturbances of people's lives during the pandemic. Nevertheless, we believe that the sample size, sampling frame, longitudinal nature and frequency of data collection, as well as our analytical approach and robustness checks, are major strengths of this study. In summary, our investigation provides a thorough contribution to understanding the dynamics of population health during the COVID-19 pandemic.

While sleep quality can sometimes be treated as a personal lifestyle factor and individual responsibility, our research identified a number of factors that contributed to sleep quality in the first 16 months of the pandemic which are dictated in parts by public health policy and health promotion. Additionally, we identified inequalities in sleep quality based on age, gender, and parenthood that would require targeted policy attention.

Notably, we show that the high mortality rates not only drove a deterioration in sleep quality directly in the five European countries we studied, but that important mediating factors exist that policy-makers should address. First, we show that part of the adverse sleep quality effects of the pandemic are moderated by the amount of time spent outside. However, outdoor time was significantly reduced when individual movement and social lives were heavily restricted during lockdowns to lower the chance of infection. To prepare for potential future surges or other pandemics, long-term strategies should be in place that enable and encourage people to maintain a certain level of autonomy over their lifestyles while preserving safety levels needed to stop viral transmission.

Second, mental health was found to be an important mediator of the effect of higher pandemic death rates on sleep in this study, consistent with the larger deterioration in sleep quality for those with worse mental health found by [65] in Italy in the early stage of the pandemic (March 2020). Given the rapid increase in mental health problems during the pandemic [8, 66], the COVID-19 pandemic and spread of mental ill-health needs to be considered a dual health crisis that requires urgent action and systemic response. Acknowledging the potential of future pandemics, as well as the current energy crisis, threats of war and climate change disasters, it is unlikely that mental health needs will become less relevant in future years. Accordingly, an easing of the pandemic should not be expected to lower mental health needs, and public health policy-making should prioritise a long-term mental health strategy.

Third, we found that the relationship between higher death rates and poorer sleep was partly mediated by coronaphobia and low trust into governments' pandemic responses. Previous studies suggest that public messages need to be trustworthy and disseminated consistently so that the population at large knows how to access, understand and interpret information, independent of educational background and health literacy, avoiding information overload and without the repeated fear mongering that was commonly felt during the pandemic [67–69]. These widely acknowledged problems have resulted in several suggestions of how public communication and government trust could be improved, including more public dialogue and greater transparency in how governance decisions are taken, and mutual decision-making with citizen committees and population representatives [68, 70]. Proposed strategies also take on lessons from previous global health crises, such as the involvement of community leaders other than government and health officials, and translation of information into minority languages, such as was the case during this pandemic in Norway [67]. Overall, our findings highlight the important role of comprehensive public health strategies in response to the complexities of the COVID-19 pandemic that relate to sleep quality, and which would likely impact individual well-being in future health crises.

## Supporting information

**S1 Fig. Survey timeline and participation rate.** *Notes*: The figure refers to all survey respondents in COME-HERE. Each histogram bar represents the number of respondents in a given day (y-axis on the left). Blue dots indicate the number of total respondents per wave (y-axis on the right). (TIF)

**S1 Table. Sleep quality and COVID-19 deaths–full pooled and panel results.** *Notes*: These are linear regressions. The sample here is respondents coming from the four 2020 waves and the first two 2021 waves of the COME-HERE survey. All the continuous variables are standardised over the estimation sample. Standard errors in parentheses are clustered at the individual level. All regressions include wave and country fixed-effects *, **, and *** respectively indicate significance levels of 10%, 5% and 1%. (DOCX)

**S2 Table. Sleep quality and COVID-19 deaths–robustness checks.** *Notes*: The sample here is respondents coming from the four 2020 waves and the first two 2021 waves of the COME-HERE survey. The dependent variable is the continuous sleep quality, except in column (3) where the dependent variable is a dummy equal to one for high sleep quality (sleep quality above the median). The continuous sleep quality and the average daily deaths variables are standardised over the estimation sample. Standard errors in parentheses are clustered at the individual level, except in column (5) where they are clustered at the four-week average daily deaths per 100,000 inhabitants' level. These are linear regressions, except in column (2) where we used the BUC estimator [60]. Longitudinal and cross-sectional weights were respectively used in columns (6) and (7). The individual controls are age categories, gender, education, parenthood, relationship status, and population density (all measured at Wave 1), the log of equivalent household disposable income in PPP, and a dummy for the employment status. The pandemic policies are the two-week averages of the Stringency Index and Economic Support Index. All regressions include wave and country fixed-effects *, **, and *** respectively indicate significance levels of 10%, 5% and 1%.
(DOCX)

**S3 Table. Sleep quality and COVID-19 deaths–heterogeneity analysis.** *Notes*: These are linear regressions. The sample here is respondents coming from the four 2020 waves and the first two 2021 waves of the COME-HERE survey. Sleep quality and the average daily deaths variable are standardised over the estimation sample. Standard errors in parentheses are clustered at the individual level. The individual controls are age categories, gender, education, parenthood, relationship status, and population density (all measured at Wave 1), the log of equivalent household disposable income in PPP, and a dummy for the employment status. The pandemic policies are the two-week averages of the Stringency Index and Economic Support Index. All regressions include wave and country fixed-effects *, **, and *** respectively indicate significance levels of 10%, 5% and 1%.
(DOCX)

**S4 Table. Mediators and COVID-19 deaths–panel results.** *Notes*: These are linear regressions. The sample here is respondents coming from the four 2020 waves and the first two 2021 waves of the COME-HERE survey. The average daily deaths variable are standardised over the estimation sample. Standard errors in parentheses are clustered at the individual level. The individual controls are age categories, gender, education, parenthood, relationship status, and population density (all measured at Wave 1), the log of equivalent household disposable income in PPP, and a dummy for the employment status. The pandemic policies are the two-week averages of the Stringency Index and Economic Support Index. All regressions include wave and country fixed-effects *, **, and *** respectively indicate significance levels of 10%, 5% and 1%.
(DOCX)

## Acknowledgments

We would like to thank Nick Adnett, Liyousew Borga, Conchita D'Ambrosio, and Rémi Yin for their help and advice.

## Author Contributions

**Conceptualization:** Anthony Lepinteur, Alan Piper, Claus Vögele.

**Data curation:** Giorgia Menta.

**Formal analysis:** Anthony Lepinteur.

**Funding acquisition:** Claus Vögele.

**Software:** Anthony Lepinteur, Giorgia Menta.

**Writing – original draft:** Samira Barbara Jabakhanji, Anthony Lepinteur, Giorgia Menta, Alan Piper.

**Writing – review & editing:** Samira Barbara Jabakhanji, Anthony Lepinteur, Giorgia Menta, Alan Piper, Claus Vögele.

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
