## [Decision Letter · Decision Letter 0]

17 Nov 2022

PONE-D-22-28393Sleep Quality and the Evolution of the COVID-19 Pandemic in Five European CountriesPLOS ONE

Dear Dr. Menta,

Thank you for submitting your manuscript to PLOS ONE. After careful consideration, we feel that it has merit but does not fully meet PLOS ONE’s publication criteria as it currently stands. Therefore, we invite you to submit a revised version of the manuscript that addresses the points raised during the review process.

We look forward to receiving your revised manuscript.

Kind regards,

Dr. Gayathri Delanerolle

Academic Editor

PLOS ONE

Journal Requirements:

2. Please provide additional details regarding participant consent. In the ethics statement in the Methods and online submission information, please ensure that you have specified:

(1) whether consent was informed;

(2) what type you obtained (for instance, written or verbal, and if verbal, how it was documented and witnessed). 

If your study included minors, state whether you obtained consent from parents or guardians. If the need for consent was waived by the ethics committee, please include this information.

"Financial support from the André Losch Fondation, Art2Cure, Cargolux,CINVEN Foundation and COVID-19 Foundation, under the aegis of the Fondation de Luxembourg, Fonds National de la Recherche Luxembourg (14840950 –COME-HERE) and the French Agence Nationale de la Recherche (ANR-21-CO16-0002) is gratefully acknowledged."

"Data collection was possible thanks to funding from the André Losch Fondation (https://www.loschfondation.lu), Art2Cure (https://www.bil.com/galerie-lindependance/art2cure/index.html), Cargolux (https://www.cargolux.com/), CINVEN Foundation (https://www.cinven.com/) and COVID-19 Foundation (https://www.fdlux.lu/en/node/1643), under the aegis of the Fondation de Luxembourg, Fonds National de la Recherche Luxembourg (14840950 –COME-HERE, https://www.fnr.lu/) and the French Agence Nationale de la Recherche (ANR-21-CO16-0002, https://anr.fr/projet-ANR-21-CO16-0002).

Reviewers' comments:

Reviewer's Responses to Questions

**Comments to the Author**

1. Is the manuscript technically sound, and do the data support the conclusions?

Reviewer #1: Yes

2. Has the statistical analysis been performed appropriately and rigorously? 

Reviewer #1: Yes

3. Have the authors made all data underlying the findings in their manuscript fully available?

Reviewer #1: Yes

4. Is the manuscript presented in an intelligible fashion and written in standard English?

Reviewer #1: Yes

5. Review Comments to the Author

Reviewer #1: The paper is well written and surely deserves publication in Plos One.

Literature is complete and recent.

The econometric part has been performed with accuracy; the panel design is appropriate and also the adopted controls are correct.

I have only few concerns. Recent literature often considers the size of a municipality (or population density of the municipality of residence) as an important control in order to split urban areas and rural ones.

My suggestion is to control for this covariate.

In addition, I don't se the estimates for the respondents' education level for every level, but I can see also a dichotomization (low level vs. post-secondary) Is it correct? I would like to see more stratification and an homogeneity in the label of the education categories.

6. PLOS authors have the option to publish the peer review history of their article (what does this mean?). If published, this will include your full peer review and any attached files.

Reviewer #1: **Yes: **Emiliano Sironi

---

## [Author Response · Author response to Decision Letter 0]

21 Nov 2022

We thank the reviewer for their comments. Our detailed response to the points raised can be find in the attached file "Response to Reviewer".

---

## [Editor Report · Decision Letter 1]

28 Nov 2022

Sleep Quality and the Evolution of the COVID-19 Pandemic in Five European Countries

PONE-D-22-28393R1

Dear Dr  Menta,

We’re pleased to inform you that your manuscript has been judged scientifically following the revisions completed and is now suitable for publication. The manuscript will be formally accepted for publication once it meets all outstanding technical requirements.

Kind regards,

Dr Gayathri Delanerolle

Academic Editor

PLOS ONE

---

## [Editor Report · Acceptance letter]

6 Dec 2022

PONE-D-22-28393R1 

Sleep Quality and the Evolution of the COVID-19 Pandemic in Five European Countries 

Dear Dr. Menta:

I'm pleased to inform you that your manuscript has been deemed suitable for publication in PLOS ONE. Congratulations! Your manuscript is now with our production department. 

Kind regards, 

on behalf of

Dr. Gayathri Delanerolle 

Academic Editor

PLOS ONE